# Does the Mark-to-Model Fair Value Measure Make Assets Impairment Noisy?: A Literature Review

**Tadeusz Dudycz \*** and **Jadwiga Praźników**

Faculty of Computer Science and Management, Wrocław University of Science and Technology, Wyb. Wyspiańskiego 27, 50-370 Wrocław, Poland; wydz.inf.zarz@pwr.wroc.pl
\* Correspondence: tadeusz.dudycz@pwr.edu.pl; Tel.: +48-71-320-3504

**Abstract:** With the purpose of reporting high-quality, transparent, and comparable information in financial statements, there is a strong, visible trend towards the implementation and use of International Financial Reporting Standards (IFRS), which represent the Anglo-American accounting model. According to IFRS, the fair value has become a dominant measurement paradigm. The purpose of this paper is to examine the implications of the implementation of the mark-to-model fair value measures for asset impairment tests on the relevance and reliability of information presented in financial reports. Among the three levels of the fair value hierarchy, mark-to-model is most controversial because it is susceptible to manipulation and has poor verifiability. After a systematic literature review and a synthesis of high-quality contributions in this field, we conclude that the implementation of asset impairment tests, that use the mark-to-model fair value measures, is not promising for increasing the quality and reliability of the information presented in financial statements. Unfortunately, research has shown that companies are using that tool to manage their earnings and promote managers' unethical behaviour. Furthermore, capital markets' reaction to asset impairment announcements is negative. Performed analysis can provide valuable pointers for standard setters, accounting policy makers, and researchers.

**Keywords:** IFRS; asset impairment; earnings management; accounting models; asset write-offs; fair value; mark-to-model

## 1. Introduction

As suggested by Coase [1], whenever an economic theory attempts to discover the most effective way to organise business operations, the technical tools of their implementation depend on the accounting. As highlighted by Nobes [2], there are two classes of accounting models. The Anglo-American model is adopted in countries with a strong capital market and orientation towards external shareholders. On the other hand, the Continental (European) model focuses on creditors' and other stakeholders' information requirements, especially those of tax authorities. The realisation of this concept may be recognised mainly in countries with a weak capital market. The aim of the Anglo-American model is to inform equity investors and allow discretion in the preparation of financial reports, as far as the resulting statement provides the "true and fair value". Conversely, the Continental model concentrates on the creditors and requires highly codified reporting [3]. The Anglo-American model is neutral, whereas the Continental model is prudent and focuses on preventing assets' over-valuation by setting the book value higher than the market value.

Accounting has also been influenced by the debate about the benefits and consequences of principles-based versus rules-based accounting standards conducted for decades [4]. Principles-based accounting standards are characterized by a clear declaration of intent but do not provide detailed guidelines for implementation. By contrast, rules-based standards provide more details

on implementation and compliance. Proponents of principles-based standards, however, agree that they create space for various uses, but this space is reduced when the assessment is made by specialists who correctly interpret the intentions of the standards. By contrast, proponents of rules-based standards say that they reduce the diversity of use, which increases the consistency and comparability of financial statements. Opponents, in turn, emphasize that they tend to follow the letter of the rules, losing its spirit [5].

There is a visible trend nowadays towards implementing and using International Financial Reporting Standards (IFRS), an approach that represents the Anglo-American mentality [6]. However, IFRS are considered as principles-based standards, while US Generally Accepted Accounting Principles (GAAP) are rules-based [5]. A slightly different opinion is presented by Schipper [7] who thinks that both US GAAP and IFRS belong to principles-based standards. The greatest success of the International Accounting Standards Board (IASB), which is the issuer of IFRS, occurred in 2005, when Regulation No. 1606/2002 [8] was adopted by the Parliament and the Council of the European Union (EU), dictating mandatory use of the IFRS by all companies listed in the EU, for the preparation of their consolidated financial statements from 01 January 2005 onwards. Furthermore, many other countries replaced their national standards with IFRS for some or all of the domestic companies. Today, 144 jurisdictions require IFRS for all or most domestic publicly accountable entities in their capital markets [9]. There were a number of expected advantages attached to this widespread IFRS adoption, including more efficient cross-border transactions, enhanced information transfer of financial reports (increased transparency), greater inter-company comparability of financial data, better asset prices (efficiency), lower cost of capital, and balance sheets that facilitate more efficient contracting between companies and lenders [10]. Optimism and the scale of the expectations from the IFRS implementation were well portrayed by Damant [11] (p. 29), who noted that, '*The impact will be enormous. Even before we come to positive advances, many billions will be saved by the fact that scarce capital is no longer invested in the wrong places at the wrong prices. And this benefit can spread throughout a country and internationally, producing wealth which can benefit even the poorest people and the poorest countries. It is hard to imagine any other technical device, in the hands of very few specialist professionals, which could have such a widespread beneficial effect on the world as a whole.*' However, in 2005, Ball [12] (p. 24) warned that, '*Internationally uniform accounting rules are a leap of faith, untested by experience or by a significant body of academic results*'. The greatest consequence of popularising the Anglo-American mentality through IFRS was substituting cost-based measures with market-based measures. The fair value has been seen as the dominant paradigm for measuring financial instruments and is also increasingly used to measure non-financial items [13]. A clear increase in the use of fair value measures to test the impairment of assets had a place after the financial crisis in 2008-2009. Although, Christensen and Nikolaev [14]—examining financial statements of British and German companies from 2005 to 2006—found a very limited use of fair-value accounting. While after the crisis a radical increase of the fair-value measure to test assets impairment use was noted [15]. For example, 16 European power and utility companies had more than €30 billion of assets and goodwill impaired between the years 2010 and 2012 [16]. The fair value paradigm rests on the decision usefulness paradigm [13]. IFRS 13 creates a hierarchy of inputs into fair value measurements, from the most to the least reliable, as follows:

*Level 1 inputs are the quoted prices in active markets for identical assets or liabilities that the entity can access at the measurement date. ......*

*Level 2 inputs are the inputs other than quoted market prices included within Level 1, that are observable for the asset or liability, either directly or indirectly. .......*

*Level 3 inputs are the unobservable inputs for the asset or liability*' [17] (pp. 10–11).

Thus, market prices represent the best estimate of the fair value, so long as the market conditions satisfy the fair value definition. Ball [12] agrees that the fair values contain more information than historical costs but only if one of the following conditions exists:

1.　Observable market prices that managers cannot materially influence due to less than perfect market liquidity.

2.　　Independently observable, accurate estimates of the liquid market prices.

Therefore, fair value will promote the relevant accounting numbers only in countries with well-developed capital markets characterised by high liquidity and the necessary information available for the fair value measurement [18]. However, the implementation of fair value in weak markets is more likely to increase unreliable information and noise in financial information streams. Additionally, if the market is illiquid, estimating the fair value is more likely to provide an opportunity for managers to manage and manipulate the earnings [19]. In addition, although IFRSs are applied to principles-based standards and are characterized by many possibilities of interpretation, they are to some extent rules-based [5]. As reported by Leuz et al. [20], accounting rules and how well they are enforced have a crucial impact on the properties of reported earnings. Accounting rules probably reflect the country's legal and institutional framework. According to Leuz et al. [20], companies in continental Europe manipulate profit more than Anglo-American countries. However, the most problematic and controversial aspect is the estimation of fair value on the basis of inputs, where prices cannot be observed on the market. This applies mainly to the valuation of the non-financial positions for which fair value is estimated, using mark-to-model approaches [13]. In this case, fair value accounting becomes mark-to-model accounting and the firms report only estimates of the market prices, rather than the actual market prices. This introduces 'model noise', due to imperfect pricing models and imperfect estimates of model parameters. Mark-to-model accounting also increases opportunities for managers to undertake financial manipulations, as they can influence both, the choice of models and the parameter estimates [12].

Consequently, the main premises of IFRS popularisation may not be achieved, as the base goal of the fair value concept is to support investors in their investment decisions by showing information more accurately, in order to measure the amount, timing, and uncertainty of (the prospects for) future net cash inflows to the entity. This allows them to assess their expectations about returns from an investment in equity and debt instruments [21]. Even where the mark-to-model approach applies, in accordance with IAS 36 and IAS 38, there is a requirement to periodically review long-term tangible and intangible assets in terms of possible impairment to fair value. However, the proper functioning of this approach will have a significant impact on the quality of information presented in the financial statements. Given this context, the main aim of the paper is to analyse the current state-of-the-art, concerning the influence that the mark-to-model fair value accounting has on IFRS asset impairment tests, particularly in terms of the quality, comparability, and perception of information presented in financial statements.

To achieve that purpose, a systematic literature review was performed with the following: The final data sample, covering 46 papers published in journals, contained an impact factor, including the history of the impairment standards' development and associated research during the research period from 1996 to 2016. The whole article is organised as follows: the review approach shows the selection criteria and quality thresholds of the asset impairment literature. It also includes an overview of the publications examined during the research period, as well as the data sample characteristics—by aggregation—of the articles on the main topics and the methodology used. In the asset impairment state-of-the-art review, we present papers that focus on the main topics of the collected literature, and the main findings section presents a knowledge development synthesis and the current trend of the development of research along with future directions.

## 2. Literature Review

### 2.1. Selection of Articles

A review was performed to collect the necessary literature and involved searching for materials relevant to the subject of asset impairment in the following databases: Web of Science, Scopus, EBSCO, Research Gate, Google Scholar, JSTOR, and SpringerLink, as displayed in Figure 1. The planning of the research approach began with the preparation of the research questions provided below:

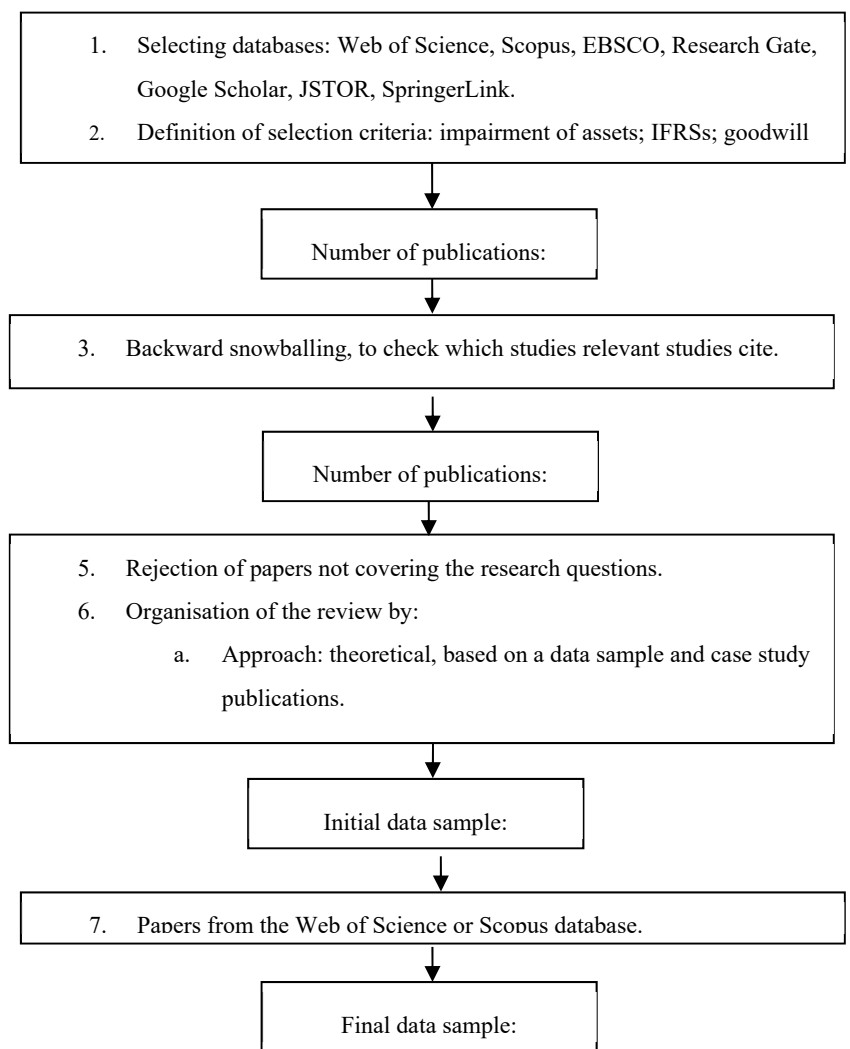

**Figure 1.** Systematic research approach.

RQ1: Which areas are considered in publications associated with asset impairment?

RQ2: What is the current trend of the development directions for asset impairment?

Based on the research questions, the definitions of selection criteria were provided, after which the preliminary research resulted in 60 publications. Then, we decided to implement backward and forward snowballing to expand the data sample and review potential publications associated with the subject of research [22]. Thus, we identified a sample consisting of 138 publications. Next, during the review of the data, we rejected publications that did not fully cover the research questions and excluded 40 articles. The last step was to introduce a quality threshold, so we decided to keep only those articles in the sample that were available in the Web of Science or the Scopus database, which resulted in 46 publications.

*2.2. Overview of the Research on Asset Impairment*

There has been a visible increase in the number of scientific articles that focus on the topic of asset impairment, since the first papers found in the mentioned databases, which were published in 1996. In Figure 2, three big peaks can be recognised in the years 2011, 2013, and 2015. The year 2011 can be identified as scientists' answer to the financial crisis from 2008 to 2009. Such a large slump exposed many of the weaknesses in the existing accounting standards, including the delayed recognition of credit losses. Also, in November 2009, the IASB proposed an impairment model in an Exposure Draft, based on expected losses rather than incurred losses. In 2012, Ramanna and Watts [23] provided

evidence of the use of unverifiable estimates in required goodwill impairment. That paper mentioned the case of the quality of write-off disclosure, along with an investigation into whether firms with the ability and motives to manage SFAS 142 on goodwill impairment losses actually do so. As of 10 November 2017, 315 other sources have cited that paper, which is one of the highest number of citations in the data sample and resulted in an increase of papers in 2013. Besides, in March 2013, the IASB presented a new Exposure Draft to recognise credit losses on a timelier basis. The final proposition of the IFRS 9 standard and the date of implementation estimated for 2018 could have influenced the growth in the number of publications during. Among the main scopes of research in the papers were the quality of write-off disclosure and findings about earnings management by impairment disclosure. Out of the 9 papers published in 2015, three were published in journals with an impact factor higher than 1 point.

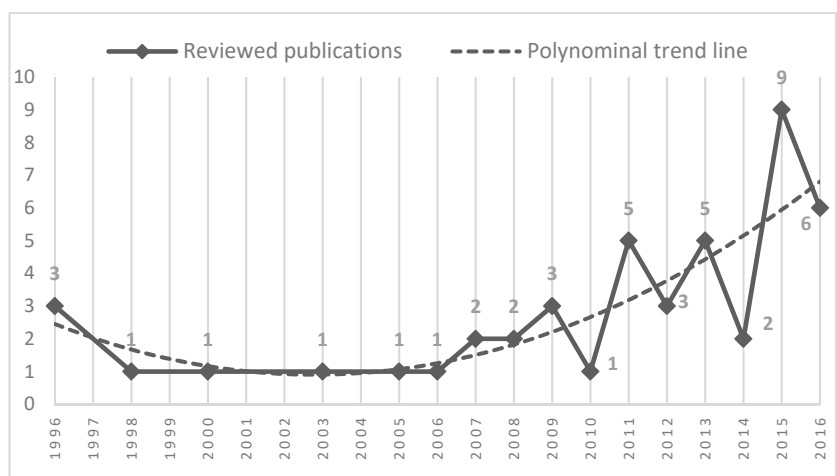

**Figure 2.** Chronological development of the number of scientific articles in the data sample, 1996–2016.

Considerable publication activity is visible in Accounting & Finance (6 articles), the Australian Accounting Review (5 articles), and the Journal of Accounting Research (4 articles), with Elliot and Hanna's [24] paper about the market reaction to the asset impairment announcement showing 555 Google Scholar citations (as of 10 November 2017).

*2.3. Data Sample Characteristics*

To identify the most important topic in the research area, the sample was grouped into fields. Table 1 presents the 4 major topics for impairment within the 15 articles presenting findings about the determinants of asset impairment. However, this subject is still being analysed to find out which financial and non-financial factors influence value impairment (e.g., Bens [25]; Rees et al. [26]; Zhuang [27]).

**Table 1.** Main topics of asset impairment research (n = 46).

| Main Topics | Number of Publications |
|---|---|
| Earnings management | 16 |
| Factors influencing write-offs | 15 |
| Market reaction to asset impairment announcements | 5 |
| Value-in-use discount rate | 4 |
| Firms' investment opportunities' influence on impairment | 2 |
| Goodwill impairment's impact on future cash flows | 2 |
| Impairment characteristics in a particular country | 1 |
| Implications for asset impairment | 1 |
| *Grand total* | 46 |

The methodology used in the reviewed publications centres on regression analyses, with a sum of 21 papers (as shown in Table 2) generally representing this particular research approach and greatly limiting the capabilities for further improvement of the methods provided. It seems to be the most representative way for researchers to measure the factors influencing write-offs. Many regression variants can be found in the data sample, for example, the ordinary least squares (OLS) regression model [28,29] and the probit model, whereby the dependent variable can only take two values to find the explanatory variables for the impairment decision [30]. To determine the impact of hypothesised drivers on the write-off decision, Garrod et al. [31] estimated four separate logistic regressions for the companies included in the total sample. Mathematical models, such as the random-forest (RF) model provided by Chen and Wu [32], have also focused on the asset impairment drivers. The diagnosis of the asset impairment factors, presented by Chen and Wu [32] used the RF model, showing a new approach to calculating asset impairment other than regression analyses. The magnitude of potential determinants, including financial information, the economic environment, and management incentives, complicates the process of verifying the asset impairment test's correctness. For the study, the authors used the RF model, suggesting that it outperforms the linear models, such as logit or tobit models, as it is more parsimonious and produces more consistent results. However, the authors admitted that the improvement shown was not as significant as expected and that there other factors still need to be included to obtain a more accurate model specification. The research sample also contained questionnaire surveys, mainly relating to the topic of management decisions regarding impairment announcements and test performance [33].

**Table 2.** Methodology used in the reviewed publications (n = 46).

| *Methodology* | *Number of Publications Reviewed* |
|---|---|
| Regression analyses | 21 |
| Descriptive analyses | 10 |
| Mathematical and simulation models | 6 |
| Discussion papers | 2 |
| Factor analyses | 2 |
| Structural equation models | 2 |
| Case studies | 1 |
| Statistics of questionnaire surveys | 1 |
| Theory papers | 1 |
| *Grand total* | 46 |

Regression analyses were a predominating methodology used in the research. After deeper investigation, we excluded 90 independent variables and recognised a few factors that were commonly used by the authors. Company size, expressed by either total assets or a logarithm of firms' sales, was the most commonly used factor. That, together with market-to-book ratio (i.e., the second most popular factor used), were mentioned by Francis et al. [34], suggesting that those measures are still adequate in terms of asset impairment factors. As shown in Figure 3, a total of 9 papers recognised goodwill as another significant independent variable. The most interesting variable was CEO, which describes a management change (particularly with the CEO position) during the research period or the last year of the CEO's tenure. Seven papers mentioned CEO as a factor to consider if there are management incentives on write-off announcements.

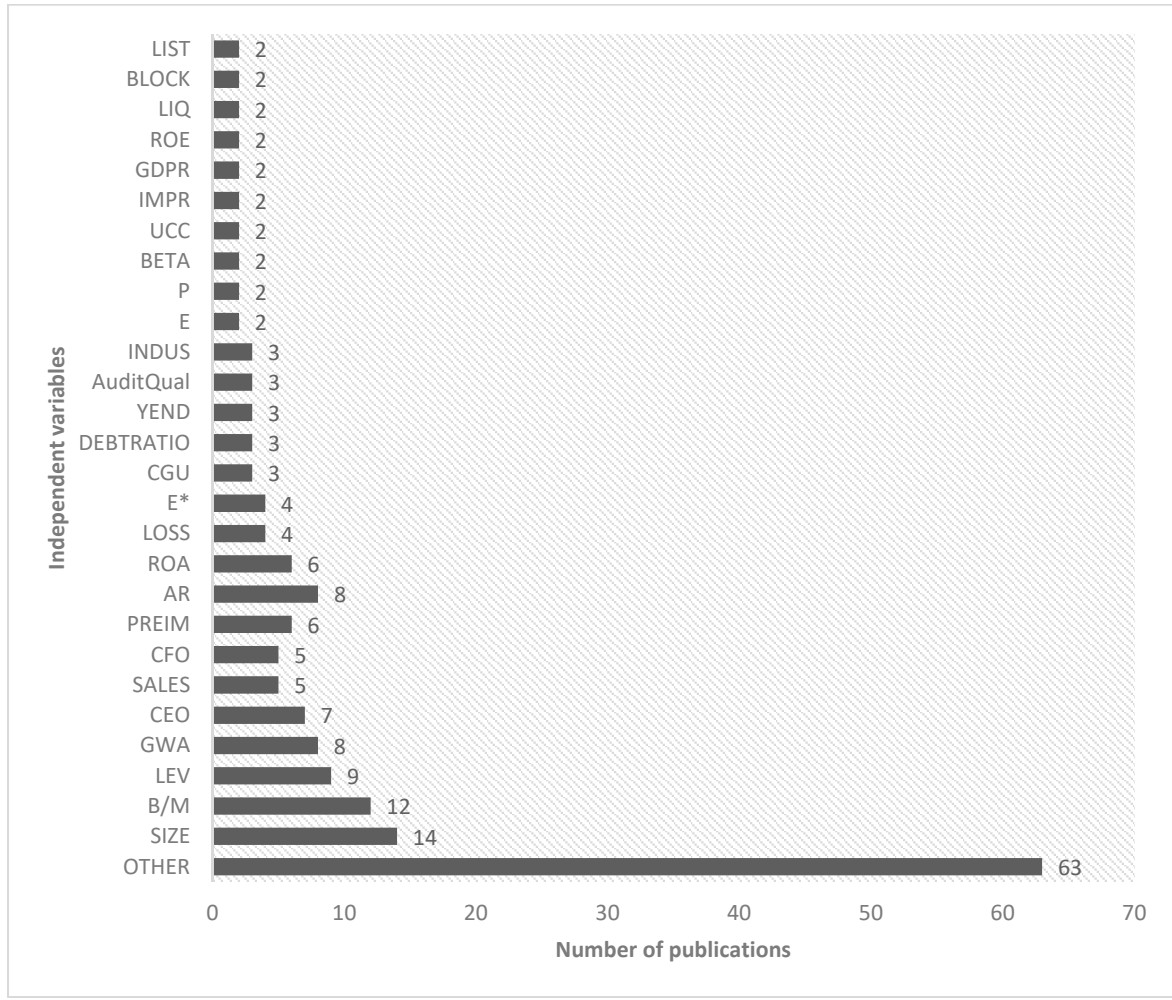

**Where:** SIZE—Company size measured as the total assets at the end of year t or calculated as a logarithm of firms sales in year t. B/M—Relationship between book to market value. LEV—Leverage measured as the total liabilities divided by total assets at the end of year t, or total debt scaled by market value. GWA—Goodwill value. CEO—Measure considered in research as a management change, new CEO in year t, or the last year of his tenure. SALES—Sales growth. CFO—Change or value of operating cash flows over the year t. PREIM—Pre-impairment earnings in year t. AR—Market return, considered as a share return of firm in year t. ROA—Return on Assets. LOSS—Absolute value of loss before reversals. E*—Accounting earnings per share after adding back the after-tax write-down amount. CGU—Measures if firm has more than one cash-generating unit. DEBTRATIO—Firms total debt divided by total assets. YEND—Measures if observation is for the post-transition regulatory change. AuditQual—Measures if the firm is audited by a one of Big 4 auditor (PwC, EY, Deloitte or KPMG). INDUS—Industry indicator. E - Reported accounting earnings per share. P—Price per share at the end of t. BETA—Beta coefficient of firm in year t. UCC—Measure of unconditional conservatism of firm in year t. IMPR—Lagged goodwill impairment loss. GDPR—GDP growth rate. ROE—Return on Equity LIQ – Measure represented by cash holding, or cash holding divided by total assets. BLOCK—The cumulative percentage of outstanding common shares held by blockholders holding and who are not part of the board of directors. LIST—Measures if company is cross listed on stock exchanges.

**Figure 3.** Independent variables used in regression analyses.

## 3. State-of-the-Art Review of Asset Impairment

### 3.1. Factors Influencing the Writing off of Asset Value

The appearance of an asset impairment scale was not examined often in the research sample. One of the first studies on the determinants of asset impairment, presented by Cotter et al. [35], showed that the average write-off was $10 million and consisted of a 0.044 part of the asset sum.

One of the most commonly reported factors influencing the write-off announcement was widely understood as the shaping of business profitability. A negative correlation of this factor with the influence of asset impairment was reported by Rees et al. [26], Giner and Pardo [30], Garrod et al. [31], Cotter et al. [35], AbuGhazaleh et al. [36], Kabir and Rahman [37], and Lapointe-Antunes et al. [38]. These researchers showed that a decline in profitability (i.e., a return of equity) influences write-offs' appearance. Lapointe-Antunes et al. [38] suggested that the recognition of goodwill impairment is caused by companies' willingness to minimise the deviation of the ROE (return of equity) and ROA (return of assets) indicators from the industry median. Laskaridou and Vazakidis [39] showed a statistically significant difference in the ROA factor between impairment and non-impairment firms, which implies that companies with lower earnings are more prone to announce write-offs and then reduce their diminished earnings (confirmed also by Godfrey et al. [40] and Yang et al. [41]). Kvaal [42] has shown that an incorrectly applied pre-tax discounting may affect small impairment recognition, due to small discount rates, which consequently do not meet the standards setters' ambition to achieve the same present value as after post-tax discounting.

A significant factor recognised by Fernandes et al. [43] and Szczesny and Valentincic [44] is the size of the company. Fernandes et al. [43] found that the probability of an impairment loss announcement is correlated positively with an affiliation to the biggest business entity and negatively with the market value. Moreover, Alciatore et al. [45] showed that a decline in asset values, reflected in returns prior to the impairment announcement, is correlated with the write-down amount. A change in management is a relevant systematic factor demonstrated in the articles. This dimension was mentioned by Bens [25], Giner and Pardo [30], Chen and Wu [32], Cotter et al. [35], AbuGhazaleh et al. [36], Lapointe-Antunes et al. [38], and Bond et al. [46]. The results in those articles suggest that the decision about impairment recording is more associated with managers' incentive to convey expectations about their good performance, rather than presenting reliable information to the stakeholders about the company's financial condition. Zhuang [27] pointed out in his paper that, when measuring the CEO changes as one of the impairment factors, researchers need to consider the nature of CEO changes and to determine whether any patterns of asset impairment emerge. On the other hand, the research sample contained two reports showing no influence in management incentives on write-off announcements. Chen and Wu [32], analysing the factors using the random-forest model, showed that more weight is placed on financial information than on the economic influence or management incentives. Kabir and Rahman [37] also indicated a weak association between contracting incentives and goodwill impairment.

A higher probability of asset write-offs, according to the accounting approach, was shown by Vogt et al. [47], who stated clearly that companies that record asset write-offs are probably more conservative in their accounting policy, as there are significantly smaller differences between net revenue and cash flows for the previous financial year for them. Companies with a more aggressive accounting policy approach usually have a larger difference between net revenue and cash flows. AbuGhazaleh et al. [36] and Kabir and Rahman [37] showed that impairments are strongly associated with the governance mechanism. The important findings of Kabir and Rahman's [37] research suggest that a strong government in a company increases the connection between economic factors and asset write-offs. However, a strong government cannot definitively eliminate illegitimate asset impairment announcements, especially when the income before the write-off is declared as negative and when the value loss occurs in the first year of a new CEO's tenure. The research results suggest that it is important to have strong corporate governance rules to ensure regime implementation of the IFRS.

### 3.2. Asset Impairment as an Opportunity for Earnings Management

Companies often resort to asset impairment in a bad financial situation, which has been proven by the negative correlation between profitability and write-off amounts in the previous section. Consequently, earnings management attempts may be suspected. The scale of these attempts may influence suspicions of earnings manipulation, that is, by the creation of financial reserves to transfer

financial results from better times to worse ones. This factor was reported quite frequently in the research.

The management's attempt to improve future earnings by writing off assets has been investigated by many researchers. Among the first were Rees et al. [26]. Based on a final data sample of 277 firms, the authors found that managers proved asset impairments when the earnings were already low in the year when the impairment was recognised in relation to the industry median. This interpretation was made after the adjustment of earnings for write-downs. They also tried to prove whether abnormal accruals taken concurrently with asset write-offs are caused by attempts to manage the company's earnings, but the results provided in the paper may not fully explain whether management considered that kind of earning potential. The authors mentioned that a purpose of the impairment can also be an attempt to improve future earnings or respond to assets' decreasing ability to generate income. Laskaridou and Vazakidis [39] used the return on assets to measure earnings management in food and beverage listed companies. They confirmed that companies with earning problems were more likely to record asset impairments and then reduce their weak results. The manipulation of goodwill impairment was shown by Alciatore et al. [45], who found that write-offs provide a value-relevant adjustment to the earnings of gas and oil companies.

Problems arise with the correctness offs write-offs, which can be explained by inconsistent regulations, as the reversal of losses in the US GAAPs is prohibited but required in the IFRS. The important topic of write-off reversals was pointed out by Chen et al. [48], who reported that, if impairment reversals are possible in the regulatory-based reporting, the quality of this information can be negatively affected. They found evidence that Chinese companies are reversing write-offs to avoid trading suspension or delisting due to the profitability-based regulation in their country, which shows that the intention of the asset impairment standard does not fully cover the desirable quality of financial reporting. Trottier [33] also confirmed that, based on 118 managers' decisions, they are more likely to record impairment when reversals are permitted. This result implies that the impairment demonstration in the financial statement in the company is still subjective and there is a poor chance of verifying managers' decisions to record impairment. Moreover, those decisions are unlikely to be questioned by the financial control in the particular country. André et al. [28] also pointed out that asset impairment actually biases the financial statement negatively because of different valuation models, managers' subjectivism, and potential manipulations caused by the unverifiable fair value estimates.

The following research focused, not only on the methods of earnings management disclosure caused by asset impairment, but also on the negative conclusion and effects of impairment announcements, such as unethical behaviour and write-off avoidance and delay. Caruso et al. [49] undertook an empirical analysis of earning management in the context of mergers and acquisitions, and they showed that, after the adoption of the IAS/IFRS, managers' behaviour changed considerably. They widely grasped the opportunity to report goodwill impairment. Moreover, the data sample limitation to 17 firms strongly limited the test, and the authors suggested that there was no earning management practices, due to asset impairment, but there was evidence of big bath, income maximisation, and income-smoothing cases. They assumed that financial reports cannot be considered reliable documents of the corporate communication to stakeholders, thereby showing their weakness. Andrews [50] pointed out that there is clearly a greater degree of big bath accounting when compared with the pre-regulatory change period before 2005 in the United Kingdom. Bond et al. [46] provided a comparison of the implementation of write-offs performed by managers of Australian firms before and after IFRS adoption, and they showed that most firms with impairment indicators still do not recognise the impairment. Avoidance of the impairment recognition was also proved by Ramanna and Watts [23], and later with an extended sample by Filip et al. [51]. Both studies identified suspected firms with booked goodwill and a market-to-book ratio (MTB) below one at the end of the fiscal year. They presented a comparison of suspected firms that did not recognise impairment with firms that carried similar goodwill and decided to report impairment in the same industry sector. Those studies

commented that avoidance is a strategy for managers to realise their motivation, according to their compensation contract, reputation, and so on. Ramanna and Watts [23], by measuring SFAS 141, provided evidence that managers avoid timely goodwill write-offs because of their interest in increasing their compensation and protecting their reputation from the implications of an asset impairment announcement. Conversely, Wang et al. [52] showed that managers report asset impairment if entities have worse performance than other firms that do not recognise impairment, which may suggest that the research results depend on the country of research. The results provided by Wang et al. [52] were obtained in Taiwan, which is an emerging market, and the studies by Filip et al. [51] and Ramanna and Watts [23] were conducted in the USA. According to the Ball [12] categorisation, Taiwan is in a legal system group significantly different than USA, which also explains international differences in financial reporting practice. Nevertheless, the results show that Common Law countries (such as the USA), identified as investor-oriented, provide information affected by the managers' interest. Ji [53] showed that evidence of avoidance is strongly visible when a company's earnings are more acutely affected by goodwill impairment loss.

Delayed goodwill impairment charges were also shown by Majid and Lode [54], who indicated that a decline in market capitalisation below the book value of the net asset is not a sufficient proxy for potential goodwill impairment. The research was undertaken in the emerging market of Malaysia. The authors argued their findings by concluding that market capitalisation cannot ideally be referred to the net assets' book value, as it cannot reflect the condition of the cash-generating units (CGU), which contain goodwill, but can be used as a starting point for the investigation of potential late reporting of goodwill impairment. Guthrie and Pang [55] examined the implication of the standard for Australian firms, which requires assets to be grouped on the lowest level, at which the cash flows are independent from the other groups of assets. Based on their research, goodwill impairment is more likely to be recognised by firms with more than three cash-generating units, which may lead to the conclusion that a higher level of asset grouping leads to a lower likelihood of the performance of asset write-offs. Thus, the proposition of CGU asset grouping can be open to potential manipulation, as the authors assumed that CGU aggregation still involves a lack of auditing. Incorrect CGU grouping, and their negative impact on overall goodwill valuation, has also been pointed out by Carlin et al. [56], Carlin et al. [57], Linnenluecke et al. [58]. Rennekamp et al. [59] proved that managers aim to maintain a positive self-image with their decisions. Giner and Pardo [30] suggested that recording asset impairment is caused by managers' unethical behaviours, derived from the avoidance of earnings surprises when they have a worse financial year than they expected. Managers want to achieve stable earnings figures from goodwill impairment recognition in relation to the past and estimated desired net income.

The research examined showed that there are no explicit signals confirming quality improvement of financial statements after the adoption of required asset impairment testing. Most of the reports signalled that the main premise of write-off announcements is earnings management. By their avoidance or performance at appropriate times, managers try to achieve benefits for the company by creating reserves for worse times in the future or to smooth the difference in earnings from the assumed plan. The quality of write-offs is imposing, especially in the papers by Andrews [50] and Karampinis and Hevas [60]. Karampinis and Hevas [60] found that the asymmetric accounting treatment of tangible and intangible impairments increases the timeliness of goodwill impairment, but negatively affects its reliability in future cash flow forecasts. Tangible assets are tested only in cases in which relevant indicators of impairment exist, and intangible assets, like goodwill, are tested annually. However, goodwill should be the first impaired asset. This treatment determines corresponding impairments of those two types of assets. As such, the announcement of goodwill impairment will be presented in time, but decreased. Andrews [50] indicated that, because of the many different methodologies used by the companies to calculate value in use, there is a problem influencing the quality of write-offs. Furthermore, there is a visible increase in the individual interpretations of the measurement of asset impairment, which is mainly because of large variations in terminology and

presentation in the companies' financial statements. Consequently, there is still a need for a debate about the systematisation of the asset impairment recognition process and the development of a methodological approach to the accounting measure, as well as the examination of individual elements and their impacts on the calculation of long-lived assets' recoverable amount.

From the shareholders' perspective, the usefulness of the information is indicated by their prediction power based on the financial statement. It can be measured by the correctness of signals about the company's future performance. Unfortunately, based on our data sample, we did not find any optimistic reports in that scope. Hayn and Hughes [61] measured the impairment indicators in acquisitions. They intended to understand whether investors can predict goodwill impairment based on financial statements, and they assumed that their ability to do so is highly limited (as also assumed by Sapkauskiene et al. [29], basically due to the quality of those statements. They also showed that the disclosure of the impairment is mainly delayed and taken after the entity has experienced a worse condition for a considerable period. On the other hand, some papers show that the goodwill impairment regime better reflects the goodwill value than amortisation [62]. Reports provided by Bostwick et al. [63] and Paugam and Ramond [64] have shown positive impact on the reliability of companies' information, by providing impairment information as an improvement in the future cash flow forecasting by stakeholders.

### 3.3. Market Reaction to Asset Impairment

Write-off announcements should improve the quality of financial statements. Nevertheless, investors are highly suspicious of this event and receive the information negatively, which may be explained by their perception of write-offs as an indication of overcapitalisation. The market reaction to earnings announcements, measured by the stock price change, was examined by Francis et al. [34] to show the information content of earnings components during the performance of large accounting write-offs. The research showed that investors respond negatively to inventory write-offs. The stock prices were much higher in the quarters with no impairment announcements in connection to prices (before write-offs) than when write-offs took place. In quarters when a write-off was performed, the stock price radically decreased. The stock returns were systematically lower in the case of companies performing asset impairment write-offs. In the same year, Elliott and Hanna [23] published their paper, in which they argued that firms with multiple write-offs might experience further reductions in their economic risk capital (ERC) because of potentially greater insecurity in the recurring earnings level. They also made an estimation of the ERC for 2761 firms that had between 0 and 4 or more asset write-offs. A similar result was achieved by Hirschey and Richardson [65] in their paper. It presented statistically significant negative abnormal returns due to write-off announcements. Li et al. [66] conducted a regression analysis of investors' reactions to asset impairment, and found that the price impact, despite significant impairment, was lower in the data sample in the period after the implementation of SFAS 142 than in the period before and the transition period. Thus, they proved that impairment loss is negatively correlated with the average growth of sales and operating profits. A negative market reaction can be explained from their perspective by the decrease in subsequent sales and operating profits, which led directly to impairment loss. Gu and Lev [67] found that overpriced shares lead to impairment based on cross-sectional analysis. They reported overpricing indicators, which can predict the occurrence and magnitude of goodwill write-offs. A strong positive correlation was shown between overpricing shares and subsequent acquisition intensity. Write-offs in the paper are considered as a by-product of the rational use of overpriced shares to acquire overvalued targets, and write-offs can be planned events based on the firm's investment strategy.

Other aspects can be considered as a market perception based on the economic practice, which may be assumed as the efficiency of adoption of the whole IFRS asset impairment tests. André et al. [28] documented an increase in the conditional conservatism of financial reporting (i.e., the book value of the net asset value should be related to their economic value, focusing on bad events rather than good ones) after the mandatory IFRS adoption in Europe in 2005, especially in countries with weak audit

quality. Even more interestingly, for firms that reported asset impairment, the conditional conservatism was less distinct than for the companies that did not recognise impairment in their statements.

## 4. Main Findings

The reviewed publications have shown the main interest areas in the asset impairment context. Table 3 presents the main findings of the reviewed articles. In the research presented in prestigious journals, we noticed a lack of unambiguous research results. The lack of such results may indicate an increase in the quality of information shown in financial statements after the adoption of asset impairment tests through more-timely information about the contemporary economic losses ("impairments") on long-term tangible and intangible assets [12], and more accurate value of the firm, or at least a fraction of it, which is the aim of the fair value accounting rules [13]. On the contrary, there are a number of reports suggesting that the management uses impairment announcements when it is widely understood that the profitability of the business is deteriorating. Many authors reported that presenting reliable information in financial statements is not a factor determining the preparation of impairment tests, and that they are rather associated with the managers' incentives to present the financial condition of the company as expected by stakeholders. For that purpose, managers also commonly delay the preparation of asset impairment tests. A number of reports indicated extensive use of asset impairment for earnings management. Authors related the exploitation of write-offs for earnings management and indicated impairment recording following the managers' unethical behaviour. Moreover, reports about the market reaction to asset impairment announcements were negative. Writers noticed that the stock price decreased after write-off disclosures, and investors reacted very suspiciously to those disclosures. The negative view was escalated by reports suggesting that the prediction of impairment is considerably limited and may even be impossible.

As shown in Table 3, the analysed research concerned impairment tests on non-financial or total assets, which means that measurements based on the model was used to measure the fair value of these assets.

Scepticism related to the mark-to-model fair value accounting has been presented by a number of authors since the beginning of its adoption. Moreover, many authors predicted problems with the application of fair value accounting in the emerging markets, due to the lack of sufficient institutional features that ensure proper compliance with the standards [19], as well as in countries without common law systems. This is due to the fact that fair value accounting represents the Anglo-American philosophy of financial reporting, which has a substantially greater propensity in recognising economic losses in a timely manner, in contrast to financial reporting in Continental Europe and Asia [68,69]. According to Ball [12], managers in common law countries have more incentives to report losses on time because these countries are characterised by comparatively deep markets and developed shareholders' rights, auditing professions, and other monitoring systems. Contrary to the common law countries, managers whose systems are less responsive to the interests of shareholders will have to change their habits under fair value accounting. The publications reviewed here do not allow us to say whether in common law countries, mark-to-model fair value accounting provides significantly better-quality information than in the other legal systems, especially in developing countries. This is because most of the reviewed articles cover only common law countries.

Figure 4 indicates that studies covering the common law countries are not only numerically dominant but are also ahead of time and, with one exception, do not include European developing countries. This may be a result of the lower level of science in these countries, as well as the lower interest of reputable journals in publishing research carried out in these countries. In Table 3, however, we see that all the negative phenomena related to the assets' impairment practice reported by the reviewed articles, as disclosed in common law countries. Therefore, if the suggestions of many authors are correct, that managers in non-common law countries are less qualified and less incentivised to correctly apply the mark-to-model fair value accounting in practice, the quality of financial reporting in these countries after the implementation of fair value accounting may be weak.

**Table 3.** Synthesis of the main findings.

| Main Topic | Factor Recognised | Country Covered by the Study | Type of Law * | Type of Assets | Publications | Journal |
|---|---|---|---|---|---|---|
| (1) | (2) | (3) | (4) | (5) | (6) | (7) |
| Factors influencing write-offs | Earnings management as the factor influenced by the asset write-offs. | UK | Com | Goodwill impairment | AbuGhazaleh et al. [36] | Journal of International Financial Management & Accounting |
| | | Australia | Com | Goodwill impairment | Cotter et al. [35] | Accounting & Finance |
| | | | | Goodwill impairment | Kabir and Rahman [37] | Journal of Contemporary Accounting & Economics |
| | | Slovenia | Cod | Fixed and current assets | Garrod et al. [31] | Journal of Business Finance & Accounting |
| | | Spanish | Cod | Goodwill impairment | Giner and Pardo [30] | Journal of Business Ethics |
| | | Canada | Com | Goodwill impairment | Lapointe-Antunes et al. [38] | Canadian Journal of Administrative Sciences |
| | | Greece | Cod | Total Assets | Laskaridou and Vazakidis [39] | Procedia Technology |
| | | US | Com | Total Assets | Rees et al. [26] | Journal of Accounting Research |
| | | | | Goodwill impairment | Godfrey et al. [40] | Accounting and Finance |
| | | China | Asia | Total Assets | Yang et al. [41] | Service Systems and Service Management |
| | Reporting of the impairment in case of worse performance than other entities that are not recognizing the write-offs. | Taiwan | Asia | Total Assets | Wang et al. [52] | International Journal of Production Research |

**Table 3.** *Cont.*

| Main Topic | Factor Recognised | Country Covered by the Study | Type of Law * | Type of Assets | Publications | Journal |
|---|---|---|---|---|---|---|
| | Size of the company as a determinant of asset impairment announcements | US | Com | Non - financial assets impairment | Alciatore et al. [45] | Journal of Business Finance & Accounting |
| | | Portugal Spain | Cod Cod | Tangible and intangible assets | Fernandes et al. [43] | Review of Business Management |
| | | Germany | Cod | Fixed and current assets | Szczesny and Valentincic [44] | Journal of Accounting and Economics |
| | Goodwill impairment may be not reflected correctly using pre-tax discounting, because of using to small discount rate and consequently reflecting to small impairment. | Europe | Cod | Long - lived, fixed assets, goodwill. | Kvaal [42] | Journal of Business Finance & Accounting |
| Earnings management | Change in management as a determinant of asset impairment announcements. | UK | Com | Goodwill impairment | AbuGhazaleh et al. [36] | Journal of International Financial Management & Accounting |
| | | US | Com | Goodwill impairment | Bens [25] | Journal of Accounting Research |
| | | Australia | Cod | Non - current assets impairment | Bond et al. [46] | Accounting & Finance |
| | | | | Total Assets | Cotter et al. [35] | Accounting & Finance |
| | | | | Non - current assets impairment | Zhuang [27] | Accounting & Finance |
| | | Taiwan | Asia | Total Assets | Chen and Wu [32] | International Journal of Information Technology |
| | | Spanish | Cod | Goodwill impairment | Giner and Pardo [30] | Journal of Business Ethics |
| | | Canada | Com | Goodwill impairment | Lapointe-Antunes et al. [38] | Canadian Journal of Administrative Sciences |
| | Strong government in a company as a factor increasing the connection between economic factors and asset write-offs. | UK | Com | Goodwill impairment | AbuGhazaleh et al. [36] | Journal of International Financial Managemen& Accounting |
| | | Australia | Com | Goodwill impairment | Kabir and Rahman [37] | Journal of Contemporary Accounting & Economics |

**Table 3.** *Cont.*

| Main Topic | Factor Recognised | Country Covered by the Study | Type of Law * | Type of Assets | Publications | Journal |
|---|---|---|---|---|---|---|
| | Asset impairment as an earnings management tool used to create reserves. | US | Com | Non - financial assets impairment | Alciatore et al. [45] | Journal of Business Finance & Accounting |
| | | | | Total Assets | Rees et al. [26] | Journal of Accounting Research |
| | | Greece | Cod | Total Assets | Laskaridou and Vazakidis [39] | Procedia Technology |
| | Asset impairment manipulation caused by impairment reversals. | China Canada | Asia Com | Long lived assets | Chen et al. [48] Trottier [33] | Journal of Accounting, Auditing & Finance Accounting Perspectives |
| | Recording impairment following managers' unethical behaviour. Misuse of the write-off opportunity. | UK Irish | Com Com | Total Assets | Andrews [50] | Procedia Economics and Finance |
| | | Italy | Cod | Goodwill impairment | Caruso et al. [49] | Journal of Intellectual Capital |
| | | Spanish | Cod | Goodwill impairment | Giner and Pardo [30] | Journal of Business Ethics |
| | | US | Com | Intangible and fixed assets | Rennekamp et al. [59] | Accounting Review |
| | Avoidance and delay of asset impairment as managers' incentives to present the financial condition of the company expected by stakeholders. | US | Com | Goodwill impairment | Filip et al. [51] | Journal of Business Finance & Accounting |
| | | | | Goodwill impairment | Ramanna and Watts [23] | Review of Accounting Studies |
| | | Australia | Com | Goodwill impairment | Ji [53] | Australian Accounting Review |
| | | Malaysia | Asia | Goodwill impairment | Majid et al. [54] | Asian Social Science |
| | | Europe | Cod | Goodwill impairment | Karampinis et al. [60] | Journal of Economic Asymmetries |
| | CGU aggregation open for potential manipulation due to lack of auditing. | Australia | Com | Goodwill impairment | Guthrie and Pang [55] | Australian Accounting Review |
| Market reaction | Negative investors' response to asset impairment announcements. | US | Com | Goodwill impairment | Gu and Lev [67] | Accounting Review |
| | | | | Total Assets | Francis et al. [34] | Journal of Accounting Research |
| | | | | Goodwill impairment | Hirschey and Richardson [65] | Financial Analysts Journal |

**Table 3.** *Cont.*

| Main Topic | Factor Recognised | Country Covered by the Study | Type of Law * | Type of Assets | Publications | Journal |
|---|---|---|---|---|---|---|
| | | | | Goodwill impairment | Li et al. [66] | Review of Accounting Studies |
| Other aspects | Unreliable information for stakeholders in the financial statements because of the low quality. | UK Irish | Com Com | Total Assets | Andrews [50] | Procedia Economics and Finance |
| | | Italy | Cod | Goodwill impairment | Caruso et al. [49] | Journal of Intellectual Capital |
| | Inability to predict impairment | US | Com | Goodwill impairment | Hayn and Hughes [61] | Journal of Accounting, Auditing & Finance |
| | | Estonia, Latvia Lithuania | Cod | Goodwill impairment | Sapkauskiene et al. [29] | Engineering Economics |
| | Impairment information as an improvement in the future cash flows value reflection. | *US* | *Com* | *Goodwill impairment* | *Bostwick et al. [63]* | *Journal of Accounting, Auditing & Finance* |
| | | Australia | Com | Goodwill impairment | Chalmers et al. [62] | Accounting & Finance |
| | | France | Cod | Total Assets | Paugam and Ramond [64] | Journal of Business Finance & Accounting |
| | An increase in the conditional conservatism of financial reporting after IFRS adoption. | Europe | Cod | Non - financial assets impairment | André et al. [28] | Journal of Business Finance & Accounting |
| | Record of the asset impairments is recognized in companies with the more conservative accounting policies. | Brazil | Cod | Goodwill impairment | Vogt et al. [47] | Revista Contabilidade & Finanças |
| | Increase of asset impairment reporting after IFRS mandatory adoption. | Serbia | Cod | Non - financial assets impairment | Andrić et al. [70] | Economic Annals |

* The term indicator: Com—Common Law countries, Cod—Code Law countries, Asia—Asian type countries.

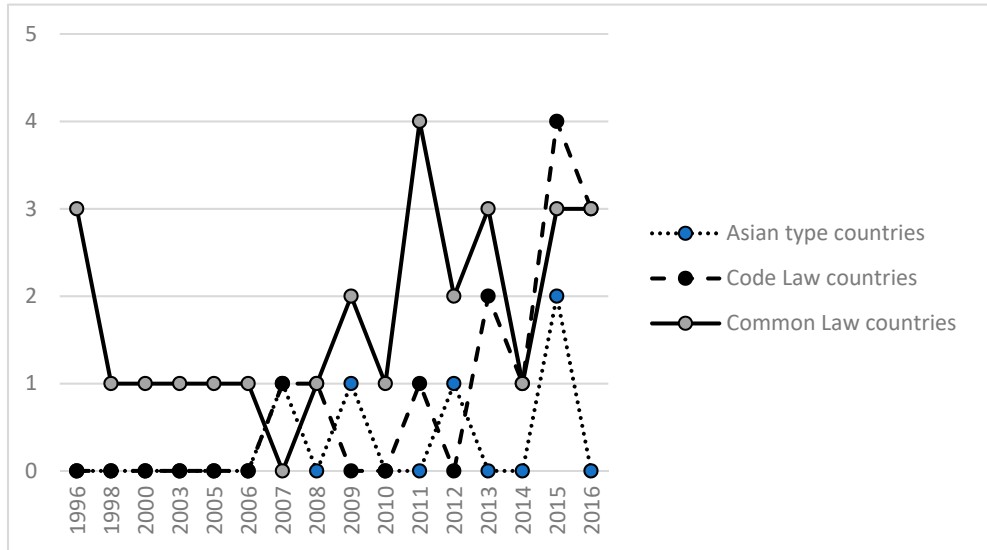

**Figure 4.** Chronological development of the number of scientific articles in the data sample based on the legal system, 1996–2016.

However, this requires further research. Discretion in the interpretation of the fair value concept, and consequently their wide use and proven application, makes the issue of standard implementation a global, rather than local, problem [10], which indicates that there is still need to improve the standard covering more specific fair value calculations guidelines. In the literature review, regardless of the visible discrepancies in the legal systems, there is still a lack of a proper notion of asset impairment tests, which is apparent in the above-mentioned cases. Transparent regulations provided by the IASB, as by Financial Accounting Standards Board (FASB), do not achieve their stated objective of increasing information value for investors. We can say, then, that the results of the research presented in the reviewed publications make highly probable the fact that the requirement to periodically review long-term tangible and intangible assets in terms of possible impairment to fair value, in accordance with IAS 36 and IAS 38, does not contribute to the primary goals of the transition from cost-based measures to fair value measures such as decision usefulness, better quality of information, and timeliness of losses. Rather, carrying out long-term asset impairment tests using the fair value model-based approach leads to obtaining noisy fair value.

## 5. Discussion

Other researchers should view the presented research results from the classical agency theory point of view. The principal in this case is the IASB, which emits IFRS and expects certain benefits in return. The agents, on the other hand, are the companies that use information asymmetry and put their own benefits above the goals expected with the use of IFRS.

IFRS implementation popularised a radically different approach to accounting reporting, involving the transition from a cost- and transaction-based model to a market-value and event-based model, which resulted in the conversion of cost-based measures into a market-based measure. The basic premise of this transition is the assumption that accounting should measure and report basic information required by the investors, which is the firm's value, or at least part of it [13]. To estimate the value of the positions that the company currently has, '*which would be received to sell an asset or paid to transfer a liability in an orderly transaction between market participants at the measurement date*' [17] (p 2), fair value measures are being applied. Fair value supporters, including the standard setters, believe that fair value measures are more useful for investors than cost-based measures [71,72]. So far, the study did not confirm these beliefs unambiguously. A review of the research, conducted by Mohammadrezaei et al. [19], shows that most empirical findings demonstrate that IFRS adoption has a positive impact on the value relevance

of accounting numbers and influences the requirement to use fair value accounting in accordance with the standard. They also show findings that IFRS resulted in less value relevance of accounting numbers. The ambiguity of these findings may be the result of the different fair value measurement levels permitted by IFRS 13 [17].

More detailed studies on the impact of fair value measures on the relevance of accounting numbers, depending on the measurement level, are shown in the results of Song et al. [73], Bosch [74], Goh et al. [75] and Kolev [76]. According to these authors, Level 3 (i.e., unobservable, business-generated inputs) fair value measurement creates less relevant information than Level 1 (i.e., observable inputs from quoted prices in active markets) and Level 2 (i.e., indirectly observable inputs from quoted prices of comparable items in active markets, identical items in inactive markets, or other market-related information), and depends on corporate governance. In companies characterised by low governance, it even creates information that is not relevant to the market valuation of the company. However, these studies were conducted for banks and financial institutions. Gassen and Schwedler [77] examined the usefulness of fair value measures from a different perspective. They surveyed professional investors and their advisers on the preferences regarding the various concepts of accounting measurement, showing that the mark-to-model fair value concept is the least useful measurement concept for investors. They also confirmed that the usefulness of the concept depends on the type of assets and liabilities. Investors assessed the mark-to-model fair value measures as the least useful for all types of assets, except financial assets for which they asses their usefulness as also low, but higher than value in use and historical cost. For investors, the fair value measures are definitely the most useful concept, provided that mark-to-market measures are used. However, when you cannot apply the mark-to-market approach, investors prefer historical cost to all types of assets, except financial assets and inventories. This means that for long-term tangible and intangible assets for which the mark-to-market approach cannot be applied, investors prefer historical costs. The results of these studies are not surprising, as the concept of fair value measures based on the model has been met with great scepticism among many researchers from the very beginning of its application. In 2005, Ball [12] predicted problems with the FASB for fair value accounting, especially with the mark-to-model concept. However, Warren Buffett said in 2002 that, in extreme cases, mark-to-model degenerates into mark-to-myth [76].

When discussing the results of the research presented in reviewed papers, we should ask a fundamental question: Does fair value, based on the model, represent decision-useful information? Seeking an answer to this question, the problem will be further analysed with the following two supplementary questions:

1.  Is financial reporting able to provide information about the company's value, which will be consistent with the market valuation of shares, or rather focus on reporting information showing a large relationship with the market valuation?
2.  Does the fair value concept allow the creation of information about the following positive properties listed by Ball [12] (p. 9)?:

    -   *'Accurate depiction of economic reality ...;*
    -   *Low capacity for managerial manipulation ...;*
    -   *Timeliness ...;*
    -   *Asymmetric timeliness (a form of conservatism): the more timely incorporation of bad news relative to good news in financial statements'.*

Regarding the first question, a problem arises at the stage of determining the detail of reporting. If we assume that reporting should reflect the market value of shares as accurately as possible, then it would be most beneficial to measure all assets together and provide one value reflecting the value of the company. However, in accordance with IFRS 13 [17], measured assets at the fair value might be either, a stand-alone asset or liability, a group of assets, a group of liabilities, or a group of assets and liabilities. Thus, there is doubt as to whether the value of the entire company will correspond to the

sum of the values of individual assets or groups of assets. Even assuming that the market is liquid, and prices are observable on the market, the sum of valuations of individual assets or groups will not cover the valuation of the whole as it will not take into account the impact of portfolio diversification or synergy. The problem is even more controversial when individual assets (groups of assets) are valued at different levels of the fair value hierarchy in accordance with IFRS 13 [17]. Therefore, we agree with Hitz [13], who states unequivocally that 'fair value accounting is neither conceived nor conceptually capable of directly measuring the value of the firm'.

In that case, what information is the most useful for investors? According to the Conceptual Framework of Financial Reporting, the answer is the one that will allow current and potential investors to make decisions to buy or hold on to equity, debt securities, or other forms of credit. The primary users also need information about the resources of the entity, not only to assess future cash flow but also to evaluate present effectivity [21]. Damant [11] (p. 30), speaking about the goals of reporting, says that '*The stewardship approach wishes to report what has been done. The decision making approach wishes the company to show figures which will enable the user to forecast what will happen*'. To analyse this problem, the assets should be divided into two basic types, depending on the place where their value is created:

1.  the assets, where the value is created on the market (e.g., financial assets, non-operating assets such as land); and
2.  the assets where value is created inside the company and depends on the unreported intangible assets.

For marketable assets whereby prices are observable on the market, a fair value measure allows you to create information that has the aforementioned features, increasing their usability for users of financial statements. They reflect economic realities correctly, they are subject to managerial manipulation on a small scale, and it is easy to report losses and profits on time. This is reflected in the preferences of investors, who strongly prefer mark-to-market fair value measures for these assets [77].

On the other hand, for assets for which value is created in a company, including long-term tangible and intangible assets, the use of mark-to-model fair value measures from the very beginning aroused many reservations, such as: high susceptibility to manipulation because managers can influence both the model and predicted parameters [12], lack of verifiability, and, as a consequence, lack of reliability [13].

If we now refer to the abovementioned goals of reporting, the mark-to-model fair value measures applied to the assets for which value is created in the company will not create useful information, both to assess what has been done and what will happen, due to the manipulation of two distinct types: Intentional and unintentional (e.g., behavioural biases). The scale and the type of intended manipulation that can be expected in reporting is shown in the results of the reviewed papers. Reporting reliable information is not a factor determining the preparation of the asset impairment tests for managers. They use these tests to manipulate the financial situation according to the expectations of the stakeholders, with some authors even suggesting that they encourage managers to behave unethically. Such behaviours occurred in common law countries, where the use of fair value mark-to-model measures did not arouse great concern among authors, due to greater motivation, tradition, and a developed institutional background. The reviewed papers, due to the small number of publications, do not allow unambiguous confirmation that such behaviours occurred outside common law countries, especially in developing countries, but it is hard to expect that the situation would be better.

Another aspect relates to unintentional manipulations resulting from behavioural biases. Behavioural finances have become a hot topic in economics in recent years. Nobel prizes have already been won, including Daniel Kahneman in 2002, Robert Shiller in 2013, and Richard Thaler in 2017, but their achievements are rarely considered in research on the quality of financial reporting. However, it should be noted that behavioural biases disclosed by behavioural economists will in many places interfere with the creation of reliable information by the fair value measures based on the model, as well as with the perception of this information by the capital market. The most typical behavioural biases listed by Barberis and Thaler [78], that may affect the quality of reported information, include:

- Overconfidence, which contributes to the overstating of their competences by managers and underestimating the risk.
- Conservatism, which contributes to the non-acceptance of information about negative changes in the enterprise by the management board and investors, will cause a stock-price drift.
- Optimism and wishful thinking, which causes the reported plans to be based on desires, to a greater extent than on the real chance of their implementation.
- Representativeness, which assess the risk of the stock markets in countries, which are not widely known to investors, as bigger than by the boards or investors from the country of the stock market itself.

Of course, this does not exhaust all the behavioural biases that can be made by both preparers of reports and users of financial reports. On the one hand, these biases will negatively affect the reliability of the information based on mark-to-model fair value measures and, on the other, its reception and trust in them. It is not surprising, therefore, that mark-to-model measures were considered the least suitable by investors [77] and, as this literature review showed, the reaction of capital markets to the disclosure of asset impairment is negative.

To assess the future cash flow, investors also need information on the competences and actions taken in the company. The carriers of this information are historical-cost measures, which are mostly in direct relation to the actions taken in the company, while fair value measures focus more on consequences. This was also reflected in the studies by Gassen and Schwedler [77], which showed that investors do not want to give up historical cost measures, but expect this information in the notes.

## 6. Conclusions

We conclude that the state-of-the-art research indicates that the implementation of asset impairment tests by the IFRS, when made by using valuation techniques from the third level of the fair value hierarchy with high probability, will not increase the quality and reliability of the information presented in financial statements.

The standard setters have to assume that, by allowing subjectivism in accounting, it will be used by managers to achieve their own goals, that are not always noble and not always in line with the objectives adopted in the standards regardless of the legal system of the country or economic development. If only there is a possibility, the practice shows (e.g., Enron or Volkswagen) that the temptation of manipulation will always overcome lofty reporting goals, especially when the situation of company deteriorates. When creating standards, one should also pay more attention to a man as a flawed being, who commits many behavioural biases affecting his activity and the results of his work

Regulation 1606/2002 of the Council of the European Union [8], which also largely contributed to the popularisation of IFRS in non-EU countries, set many goals that, in the face of the current practice of assets impairments presented in the reviewed publications, may not be accomplished for reasons detailed in Table 4.

Considering the above findings, further research should focus on the question of whether the IFRS standard has improved the degree of harmonisation of the management practice of impairment reporting (in global accounting standards). One of the future research directions should also be to gauge the asset impairment and influence of the IFRS on the decrease in information asymmetry. There is a need to focus on the impact of the IFRS on the information quality presented in the financial statements according to both the Anglo-American and Continental accounting models, with the examined research raising the following question: What kind of information are stakeholders expecting to receive in the financial statement?

**Table 4.** Council of the European Union's Regulation No. 1606/2002, regarding the asset impairment practice.

| Regulation Objectives | Asset Impairment Practice |
| --- | --- |
| A true and fair view of the financial position and performance of an enterprise. The enhancement of the transparency and comparability of financial reporting. | Non-comparable individual interpretations of the standard caused by subjective mark-to-model approach. |
| Efficient and cost-effective functioning of the capital market. The protection of investors and the maintenance of confidence in the financial markets. | Unreliable documentation of corporate communication to stakeholders because of the low quality of accounting numbers and susceptibility to manipulation. |
| Increasing convergence of the accounting standards currently used internationally with the ultimate objective of achieving a single set of global accounting standards. | Non-required disclosure of the methodology used in the asset impairment calculations. |
| Reinforcing the freedom of movement of capital in the internal market and helping to enable Community companies to compete on an equal footing for financial resources. | Market overreaction to asset impairment announcements. Not a reliable view of companies' financial condition, due to the delay in and avoidance of the impairments. |

Therefore, there is also a need to conduct further research comparing the effectiveness of Anglo-American and Continental approaches for meeting the financial market stakeholders' expectations and, as a result, to improve the quality of the signals provided to them. On that basis, further research will focus on the predictive power of the information included in financial statements, according to long-term investment decisions corresponding to signalling theory. Moreover, the research sample is limited by the methodological approach used by authors. We recognise that most of the papers were based on regression and descriptive analyses, limiting the capabilities for further improvement of the methods. This limitation also indicates a future research direction; that is, to improve the existing methods or to develop new models of the factors influencing write-off measurement.

**Author Contributions:** Initiated the idea, T.D.; funding acquisition, T.D.; resources, J.P.; writing—original draft, T.D. and J.P.; T.D. contributed to the idea development and the revision of this paper. All authors have read and agreed to the published version of the manuscript.

**Funding:** This work was supported by the National Science Centre Poland [grant number: 2017/25/B/HS4/01374].

**Conflicts of Interest:** The authors declare no conflict of interest.

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
