# Peer review of "Does the Mark-to-Model Fair Value Measure Make Assets Impairment Noisy?: A Literature Review"

_sustainability, doi:10.3390/su12041504_

Round 1

Reviewer 1 Report

This paper appropriately suggest the inherent problem in fair value accounting.

IFRS focuses on the fair value, however, the fair value cannot always be estimated or observed clearly, so that it may cause managements' descretion.

This paper especially concentrates on the impairment accounting and suggest it may confuse the users of accounting information by intensive literature review.

However, this reviewer is sorry for the following issue in this paper.

As a review paper, this paper analyzes prior studies intensively and classifies them into categories under several standards. Among them this paper applies methodology used as one standard. However, in this paper, the details of methodologies are not clearly explained. For example, the paper says 21 studies used regression analyses, but it does not suggest what the main variables are, how the main variables are estimated, and so on. So this reviewer thinks the authors had better suggest the summary of details of methodologies used in prior studies.

Thank you for submitting your precious paper and giving the chance of the review.

Author Response

The authors would like to thank the  Reviewers for their precious time and invaluable comments. We have carefully addressed all the comments. The corresponding changes and refinements made in the revised paper are summarized in our response below.

Response to Comments from Reviewer 1

Comment 1:

As a review paper, this paper analyzes prior studies intensively and classifies them into categories under several standards. Among them this paper applies methodology used as one standard. However, in this paper, the details of methodologies are not clearly explained. For example, the paper says 21 studies used regression analyses, but it does not suggest what the main variables are, how the main variables are estimated, and so on. So this reviewer thinks the authors had better suggest the summary of details of methodologies used in prior studies

Response:

There has been added graph, which represents variables used in the regression models. 90 independent variables have been excluded in papers reviewed. We also grouped variables, which were used in multiple publications, showing the same trends of assets impairment factors used in research. Further considerations of assets impairment factors are included in main body of the paper. We included also description of those variables, and how they were represented in the research.

Reviewer 2 Report

The topic is interest and the paper has some potential. I think that the flow of the paper is not well readable. Several topics are listed in a very dense text that is sometimes hard to follow.

Here there are some comments.

I suggest to refer to the debate principles versus rules accounting standards.

In the introduction you set the Continental model against the Anglo-American model. I suggest to refer also to the debate that counter the principle-based accounting standards and the rule-based accounting standards. You should refer to (Leuz et al., 2003) to incorporate the consideration of different institutional enforcement. On line 88 you wrote that "the most problematic and controversial aspect is the estimation of fair value". Later (line 101) you wrote that "the main aim of the paper is to analyze the current state of the art concerning the influence of the mark-to-model fair value accounting used to undertake IFRS asset impairment".The logic flow is not immediate because in the first part you focus on the fair value supposedely in the asset recognition. Notwithstanding the aim of the paper is focused on the impairment test. I suggest to clarify why you focus on the fair value within the impairment test instead on the fair value measurement in general.  What is your positioning referring to the empirical research about accrual measurement? I suggest to make clear from what theoretical framework you are drawing your insights while commenting the evidences. Most of the paper you cited are rely on the agency theory, is that the one you are referring to?  Table 3: column 2 is hard to understand, I suggest to review the wording to make the sentences more immediate to catch. I would add a column indicating the journals.

Author Response

The authors would like to thank the  Reviewer for their precious time and invaluable comments. We have carefully addressed all the comments. The corresponding changes and refinements made in the revised paper are summarized in our response below.

Response to Comments from Reviewer 2

Comment 1:

I suggest to refer to the debate principles versus rules accounting standards.

In the introduction you set the Continental model against the Anglo-American model. I suggest to refer also to the debate that counter the principle-based accounting standards and the rule-based accounting standards. You should refer to (Leuz et al., 2003) to incorporate the consideration of different institutional enforcement.

Response:

Differences and inconsistent regulations of US GAAP and IFRS were mentioned in section 3.2. (Asset Impairment as an Opportunity for Earnings Management). We also added to the paper consideration of IFRS, which is the principal based accounting. The standard allows a large area of interpretations and consequently the fair value concept is being differently implemented in weak and developed markets. We also added impact of effective imposed accounting rules in the companies in Continental and Anglo-American countries and how it’s being reflected in the legal and institutional framework, which was mentioned by Leutz et al. (2003).

 Comment 2:

On line 88 you wrote that "the most problematic and controversial aspect is the estimation of fair value". Later (line 101) you wrote that "the main aim of the paper is to analyze the current state of the art concerning the influence of the mark-to-model fair value accounting used to undertake IFRS asset impairment". The logic flow is not immediate because in the first part you focus on the fair value supposedely in the asset recognition.

 Response:

In terms of requirement of periodically review of long-term tangible and intangible assets, which are being tested for possible impairment to fair value, we have added an explanation that inadequate mark-to-model application have a significant impact on information asymmetry in the financial statements. That clarification has been included in the corrected paper.

Comment 3:

Notwithstanding the aim of the paper is focused on the impairment test. I suggest to clarify why you focus on the fair value within the impairment test instead on the fair value measurement in general. 

Response:

We assume that this comment is strictly associated with the previous one, and included clarification should be sufficient.

 Comment 4:

I suggest to make clear from what theoretical framework you are drawing your insights while commenting the evidences. Most of the paper you cited are rely on the agency theory, is that the one you are referring to?

 Response:

In discussion we pointed out that the presented research should be viewed from the point of view of classical agency theory.

 Comment 5:

Table 3: column 2 is hard to understand, I suggest to review the wording to make the sentences more immediate to catch. I would add a column indicating the journals.

Response:

We reviewed the “factor recognized” column and corrected some of the wording used. Column indicating the journals reviewed has been added.

Round 2

Reviewer 2 Report

Line 533: are the Author really referring to the IFSB (Islamic Financial Services Board). I was expected they refer to the IASB (International Accounting Standards Board). If this is a typo, the Author should adjust it.

I believe the Authors overlapped the conceptual reference to the rule- versus principle standards with the enforcement topic. I was suggesting to add reference to the debate rule- versus principle standards as a frame where standards leave room for discretion in different ways (like for example Schipper, K. 2003. Principles-based accounting standards. Accounting horizons, 17(1), 61-72, but there are several other more recent). Furthermore - and this is an additional reference to bring in - the institutional enforcement (Leuz et al., 2003) plays a role in how such discretion is then used by the preparers. Authors referred to the second one (institutional enforcement topic, where I explicitly suggested Leuz et al. 2003 in the previous review), while reference to the rules- versus principles accounting standard debate is not properly brought in. In my opinion Authors need to separate conceptually the two topics. Even the sentence on line 89 "according to Leuz et al. 2003, accounting rules and how they are enforced" is misleading to me. Leuz et al. 2003 discuss about accounting standards that are not always rules based (on the contrary in that paper they are discussing mostly about international accounting standards that are principles based).

Please correct Leuz instead of Leutz within the text. It is correct in the reference list.

The effort on the table 3 is appreciable, column 1 is clear, while I suggest to make more direct the labels in column 2.

Author Response

Authors’ Response to the Review 2 Comments

The authors would like to thank second time the Reviewer for their precious time and invaluable comments. We hope that this time we have carefully addressed all the comments. The corresponding changes and refinements made in the revised paper are summarized in our response below.

Comment 1:

Line 533: are the Author really referring to the IFSB (Islamic Financial Services Board). I was expected they refer to the IASB (International Accounting Standards Board). If this is a typo, the Author should adjust it.

Response:

 We have corrected this mistake.

Comment 2:

Authors referred to the second one (institutional enforcement topic, where I explicitly suggested Leuz et al. 2003 in the previous review), while reference to the rules- versus principles accounting standard debate is not properly brought in. In my opinion Authors need to separate conceptually the two topics

Response:

In introduction we referred to the debate of  the principle-based accounting standards and the rule-based accounting standards.

Comment 3:

Even the sentence on line 89 "according to Leuz et al. 2003, accounting rules and how they are enforced" is misleading to me. Leuz et al. 2003 discuss about accounting standards that are not always rules based (on the contrary in that paper they are discussing mostly about international accounting standards that are principles based).

Response:

Section has been corrected to avoid mislead and controversy.

Comment 4:

Please correct Leuz instead of Leutz within the text. It is correct in the reference list.

Response:

We have corrected this typo.

Comment 5:

The effort on the table 3 is appreciable, column 1 is clear, while I suggest to make more direct the labels in column 2.

Response:

We've tried to improve column 2 wording  to reflect the sentences more direct.